# Insight into the Relationship between Oral Microbiota and the Inflammatory Bowel Disease

**DOI:** 10.3390/microorganisms10091868

**Published:** 2022-09-19

**Authors:** Yimin Han, Boya Wang, Han Gao, Chengwei He, Rongxuan Hua, Chen Liang, Shuzi Xin, Ying Wang, Jingdong Xu

**Affiliations:** 1Department of Oral Medicine, School of Basic Medical Science, Capital Medical University, Beijing 100069, China; 2Undergraduate Student of 2018 Eight Program of Clinical Medicine, Peking University People’s Hospital, Beijing 100083, China; 3Department of Physiology and Pathophysiology, School of Basic Medical Science, Capital Medical University, Beijing 100069, China; 4Clinical Medicine, School of Basic Medical Science, Capital Medical University, Beijing 100069, China; 5Department of Dermatology, Beijing Tongren Hospital, Capital Medical University, Beijing 100069, China

**Keywords:** oral microbiota, inflammatory bowel disease, ectopic gut colonization, intestinal epithelial barrier

## Abstract

Inflammatory bowel disease has been a growing concern of lots of people globally, including both adults and children. As a chronic inflammatory disease of the intestine, even though the etiology of inflammatory bowel disease is still unclear, the available evidence from clinic observations has suggested a close association with microorganisms. The oral microbiota possesses the characteristics of a large number and abundant species, second only to the intestinal microbiota in the human body; as a result, it successfully attracts the attention of researchers. The highly diverse commensal oral microbiota is not only a normal part of the oral cavity but also has a pronounced impact on the pathophysiology of general health. Numerous studies have shown the potential associations between the oral microbiota and inflammatory bowel disease. Inflammatory bowel disease can affect the composition of the oral microbiota and lead to a range of oral pathologies. In turn, there are a variety of oral microorganisms involved in the development and progression of inflammatory bowel disease, including *Streptococcus* spp., *Fusobacterium nucleatum*, *Porphyromonas gingivalis*, *Campylobacter concisus*, *Klebsiella pneumoniae*, *Saccharibacteria* (TM7), and *Candida albicans*. Based on the above analysis, the purpose of this review is to summarize this relationship of mutual influence and give further insight into the detection of flora as a target for the diagnosis and treatment of inflammatory bowel disease to open up a novel approach in future clinical practice.

## 1. Introduction

As a lifelong illness occurring in both adults and children, inflammatory bowel disease (IBD) has been a great concern for many people globally, and seriously affects people’s quality of life. IBD can be categorized into two clinical types such as ulcerative colitis (UC) and Crohn’s disease (CD), with clinical manifestations including pus and bloody stool, abdominal pain, tenesmus, diarrhea, and weight loss [1,2]. While the incidence of IBD seems to have been stable in Western countries since 1990, global prevalence has increased in newly industrialized countries, including Asia, Africa, and South America [2,3]. Therefore, the research on IBD has never stopped, with the investigation continuing to progress and deepen the concept that IBD is the consequence of the complicated interaction of genetic, immunological, microbiological, and environmental factors [4,5,6]. However, the current knowledge of IBD and the cure problem have always puzzled scientists. Numerous studies have been conducted to gain a clear understanding of the pathogenesis and the treatment of IBD. However, the enormous potential of the oral microbiota, which contains the second largest pool of potential human pathogens, is often overlooked. Based on the above reasons, this review makes a summary of the composition and role of oral microbiota, focusing on the underlying relationship between oral microbiota and IBD, providing fresh avenues for future research directions and clinical trials of IBD.

## 2. Oral Microbiota

As the gateway to the human body, the oral cavity has a complex environment and is the habitat of complex microbial communities, including bacteria, fungi, viruses, protozoa, and archaea called oral microbiota (Figure 1) [7,8]. These commensal oral microbiota act as a guardian of oral health by restraining the colonization of opportunistic pathogens and regulating inflammatory responses [9]. However, once the equilibrium microenvironment is perturbed, it may lead to some oral ailments and gut-related diseases, even posing a threat to the overall health [4,10].

### 2.1. Characteristics of Oral Microbiota Composition

In the past, due to the limitation of traditional culture-dependent techniques, it was challenging to culture abundant oral microbiota in vitro, so the research on oral microbiota has been dramatically restricted. Commonly used molecular diagnostic methods, such as polymerase chain reaction (PCR), can only identify established targets, but cannot detect unknown microorganisms, so the composition of oral microorganism remains largely undiscovered [11]. However, in recent years, the next-generation sequencing (NGS) of hypervariable regions within the 16S rDNA of the bacterial genome has permitted us to massively improve the discrimination of bacteria [11]. Without depending on isolation and culture, the method can directly detect unknown bacteria, which significantly improves our understanding of oral bacteria. Based on the advantages of low cost and high throughput, NGS shows a broad prospect in microbial detection [11].

Within the oral microbiota, bacteria belonging to more than 700 species rank the oral microbiota as the second most diverse and most affluent microbiota of the human body [8]. In the oral microbiota of healthy adults, the bacterial phyla showing the highest abundance were Firmicutes, Proteobacteria, Bacteroidetes, and Actinobacteria, while Fusobacteria, TM7, spirochaetes, OD2, and Synergistes were less abundant phyla [12]. The oral microbiota colonizes almost any part of the oral tissue, for example, hard and soft surfaces, soft and hard palate, tongue, gingiva, buccal mucosa, teeth, and saliva. It is worth noting that the distribution of oral microbiota varies among different parts [13]. Caselli et al. collected samples from different parts of the oral cavity from twenty healthy adults and analyzed them by whole-genome sequencing (WGS) combined with qRT-PCR [11]. The expression profiles showed that *Streptococci* was the most abundant genus, especially in mucosal tissues; the *Neisseria*, *Prevotella*, and *Haemophilus* genera were also highly prevalent in most sites [11]. Bacteria belonging to the *Rothia* genus, Gram-positive round-rod-shaped bacteria belonging to the *Actinomycetaceae* family, were relatively abundant in all sites except for keratinized gingiva [11]. However, the distribution of anaerobes (*Actinomyces*, *Veillonella*, *Fusobacterium*) in the oral cavity is less, basically limited to the subgingival plaque, and *Simonsiella* was almost exclusively detected in the hard palate [11]. At the species level, *Streptococcus mitis* was the most abundant, followed by *S. oralis*, *salivarius*, and *sanguinis*, while *S. mutans* was less prevalent. In addition, *Haemophilus parainfluenzae*, *Prevotella melaninogenica*, *Neisseria subflava*, and *Rothia dentocariosa* were the most prevalent species of the respective genera [11]. The oral microbiota undergoes some changes with age. On the edentulous stage, oral microorganisms are mainly colonized on the mucosal surface, and the abundance of *Firmicutes* in the saliva is far higher than any other members of bacteria. As the deciduous teeth erupt, the colonization sites associated with hard tissues are striking incrementing due to the diversity of the oral microbiota. The abundance ratios of *Fusobacteria*, *Tenericutes*, *Synergistetes,* and *Proteobacteria* have undergone a gradually increasing trend [14]. Similarly, as the α-diversity of the salivary microbial community increases, so does the number of other species, including *Veillonella*, *Fusobacterium*, *Neisseria*, *Prevotella*, *Rothia*, *Actinomyces,* and *Corynebacterium* [15]. In addition, many studies have showed that diseases of pregnancy, intrauterine environment changes, and other vital events have a meaningful impact on the short-term and long-term health of newborns, involving, for instance, cardiovascular [16], metabolic [17], and neuropsychiatric diseases [18]. It is worth exploring whether IBD has a fetal origin. One study detected that the prevalence of periodontal pathogens was higher in the placentas of women with periodontitis than those without periodontitis, especially *Fusobacterium nucleatum* and *Eikenella corrodens* [19]. In early pregnancy, periodontal pathogens could reach the placenta by specific routes and colonize the placenta, which may lead to adverse pregnancy outcomes such as premature delivery, amnionitis, and stillbirth [20]. However, whether the development of IBD in children is related to the changes of oral microorganisms in mothers deserves further study.

Unlike the aforementioned oral bacteria, the Candidate Phyla Radiation (CPR) group of bacterial organisms has just recently come to the attention of researchers [21]. Still poorly known, these microbes were first described in the environment in 1981 as “ultramicrobacteria” with an ultra-small cell size, and were first associated with the human oral microbiota in 2007 [22]. Among more than 35 CPR phyla, TM7 receives the most attention and can be studied more extensively due to its close relationship with a variety of mucosal diseases such as vaginosis, IBD, and periodontitis [23,24,25]. As the first among the whole CPR group to be successfully cultured and stably maintained in vitro, *Nanosynbacter lyticus* type strain TM7x (detailed seen Box 1) lives on the surface of *Actinomyces odontolyticus actinosynbacter* strain XH001, which is the host bacteria, and this autoeciousness is dynamic, presumably influenced by environmental conditions including nutrient availability and oxygen concentration [26,27,28]. Perhaps the parasitic relationship between CPR such as TM7 and other oral microorganisms indirectly impacts the composition of the oral microbiota and thus influences the function of oral microbiota and leads to the occurrence of diseases, which implies that research on the CPR would be valuable in future clinical practice.

Box 1Characteristics of the TM7x.Due to its ultrasmall size (200–300 nm), TM7x is unusual among all bacteria. Settling on the surface of XH001 [28], which is the host bacterium and related to severe periodontal disease (PD) and other inflammatory conditions, is currently receiving much attention due to its unique properties. Carrying highly reduced genes (about 700 genes) limits its capability to produce amino acids, membrane lipids, and nucleotides [26,28]; however, Orth et al. have confirmed that XH001 likely provides lactate as the primary carbohydrate source for TM7x via the flux balance model [29]. In in vitro laboratory conditions with adequate nutrition, TM7x can enjoy a long-term stable relationship with XH001 [30]. TM7x, although significantly inhibiting host development and cell division, did not affect uninfected XH001, supporting another infection cycle by TM7x via horizontal transmission [30]. Interestingly, TM7x utilizes its ability to facilitate biofilm formation possibly to benefit its host cells, thereby blocking the recognition of XH001 by the activation of tissue macrophages, or possibly in a direct manner, and resides in macrophages, inhibiting TNF-α production [26,31].

Despite being an omnipresent component of the microbiota in mammals, fungi have received little attention due to their relative scarcity (0.1% of the microbiota, measured in colony-forming units) and has remained uncharacterized due to difficulties in cultivating and nomenclature [32]. Nevertheless, around half of individuals carry *Candida* species, which may be related to a variety of acute and chronic infections [33]. In a clinical cohort study, it was found that 85 fungi were found in the oral cavity through the analysis of the characteristics of oral fungi in 20 healthy subjects, and the dominating genera of fungi in the oral cavity were still *Candida*, *Cladosporium*, *Aureobasidium*, *Saccharomycetales*, *Aspergillus*, *Fusarium*, and *Cryptococcus* [34]. Furthermore, *Malassezia* is also an abundant fungal genus in a healthy oral environment, following Depuy’s study [35]. Excellent investigation has given direct evidence that the oral fungi may be implicated in several illnesses, such as inflammatory bowel syndrome (IBS), CD, chronic respiratory diseases, and hepatitis B [25]. In addition, the interplay in bacterial–fungal interactions and some disorders has become a growing focus. Fungal species obtain the ability to set up a structural “skeleton” for bacterial–fungal multispecies biofilms due to the larger cell size and the competence to create filamentous hyphae [25]. Several lines of evidence support the crucial influence of bacterial–fungal interactions in a robust host immune system [36]. The primarily study by Wu et al. cultured *Candida albicans* SN152 alone and with *Fusobacterium nucleatum* ATCC 23,726 in fetal bovine serum, as well as separate cultures of two microorganisms using a two-chamber vessel divided by a membrane. The results showed a 10-fold increase in *C. albicans* hyphal after 4 h when grown in monoculture [37]. On the contrary, there is no noticeable growth when co-culturing with *F. nucleatum* over an equal time. Interestingly, while the two were separated by a membrane, the inhibition disappeared, suggesting that *F. nucleatum* had a minimal effect on the development and mycelian morphogenesis of *Candida. albicans* in order to promote *Candida* survival through a contact-dependent pattern [37]. A further study [38] using the murine macrophage-derived RAW 264.7 cell line and in the ELISA assay proved that the production of MCP-1 and TNF-α due to the stimulation of *F. nucleatum* in macrophages can be restrained by the yeast form of *C. albicans* [38]. The significance of the evidence is not only to confirm that the interplay between *C. albicans* and *F. nucleatum* probably contributes to establishing a long-term coexistence in the oral cavity but also to reveal the importance of further understanding bacterial–fungal interactions, which may lead to new and daunting ideas for clinical treatments and laboratory investigation.

Moreover, a range of viruses, including some ordinary eukaryotic families, *Herpesviriade*, *Papillomaviridae*, *Anelloviridae,* and *bacteriophage* were found in the oral cavity, as a “barometer” for certain diseases [39]. Complementary laboratory experiments have also proved the presence of two protozoon species of *Entamoeba gingivalis* and *Trichomonas tenax* and three archaea species of *Methanobrevibacter oralis*, and two un-named *Methanobrevibacter* phylotypes [40].

### 2.2. Effects of Resident Oral Microbiota

As mentioned, highly diverse commensal oral microbiota is not only a normal part of the oral cavity but also has a pronounced impact on the pathophysiology of systemic health [40]. The presence of commensal oral microbiota can protect the host against colonization of extrinsic bacteria by reducing binding sites, and nutrient support, thereby avoiding an inflammatory state, especially against some opportunistic pathogens, including *Candida* species and *Staphylococcus aureus* [40]. Bacteriocins produced by *Streptococcus dentisani* suppress the growth of cariogenic bacterial species [41]. As germ-free mice do not have lymphoid follicles and mucosal IgA is produced only in the presence of the microbiota, oral microbes may also participate in constructing the immune barrier [40,42]. Surprisingly, oral microorganisms are also bound up with cardiovascular health (for detail see Box 2).

Box 2Influence of metabolite-derived oral microbes on cardiovascular health.Nitrates can be actively transported by salivary glands from blood to saliva via sialin (the protein product of *SLC17A5*) transporters, and then oral microbes including *Veillonella*, *Actinomyces*, *Rothia*, *Haemophilus*, and *Neisseria* convert it to nitrite. When reaching the stomach with saliva, it is decomposed into nitric oxide (NO) in an acidic environment and released into the bloodstream [40,43,44]. As a potent vasodilator and anti-inflammatory signaling molecule, NO plays a partly vital role in maintaining vascular homeostasis [45].

On the one hand, the oral microbiota is a necessary but not sufficient requirement for the progress of some common oral diseases, for instance, dental caries, gingivitis, periodontitis, and oral lichen planus even oral cancer [10,46]. As one of the most common oral diseases, dental caries is a chronic progressive destruction of dental hard tissues caused by oral bacteria as the primary pathogen under the participation of multiple factors [47]. With a high incidence, caries afflicts people of all ages, from children to the elderly [47]. The oral microbial composition in the saliva of 21 caries-free children and 20 caries-infected children was characterized and compared by the single-molecule real-time DNA sequencing system, and the result showed that *Prevotella* spp., *Lactobacillus* spp., *Dialister* spp. as well as *Filifactor* spp. were linked to the development of the pathogenesis and progression of dental caries [48]. In addition, through 16S rDNA sequencing technology, Agnello et al. found that the level of *Streptococcus mutansc* in children with severe dental caries in early childhood was significantly higher than those without dental caries in children [49].

Similarly, a large number of studies have shown that oral flora imbalance is related to oral inflammation, and may cause systemic diseases through bacteremia [50], including cardiovascular system diseases, autoimmune diseases such as rheumatoid arthritis, gastrointestinal system diseases like colorectal cancer and IBD, neurological disorders including Alzheimer’s disease (AD), as well as endocrine system diseases such as diabetes, obesity [4]. Taking AD, a current medical research hotspot, as an example, Kamer et al. found that plasma concentrations of TNF-α and antibodies against periodontal bacteria *Porphyromonas gingivalis*, *Tannerella forsythia*, and *Actinobacillus actinomycetemcomitans* were obviously enhanced in AD patients compared to controls by ELISA technique [51]. Moreover, binary logistic models were applied to confirm the independent association of these two factors with AD, which seemed to imply that some oral microbes are likely to play a part in the pathogenesis of AD through inflammatory pathways and can be used as markers for the diagnosis of AD. Recent studies investigated the bronchoalveolar lavage fluid of patients with COVID-19 via metagenomic next-generation sequencing. They found that aside from SARS-CoVs, some oral microbiota could also be characterized, such as *Capnocytophaga gingivalis*, *Veillonella parvula*, and *Prevotella melaninogenica* [52,53], indicating that coinfection of the oral microbiota with SARS-CoV-2 can occur in the lungs of patients with COVID-19 [54]. Not content with this result, Khan et al. further clarified that when *Prevotella* proteins expression exceeds a certain limit, it will endorse viral infection and participate in numerous interactions with NF-κB via host–pathogen protein–protein interaction analysis and functional over-representation analysis [55]. The above results suggest that the disorder of oral bacteria is not only an oral disease, but also a systemic disease, which should be highly valued.

## 3. Relationship between the Oral Microbiota and IBD

Despite the environmental segregation of the mouth and gut, such as gastric acidity and antimicrobial bile acids in the duodenum, according to the report, over half of the microbial species such as *Streptococcus*, *Actinomyces*, and *Veillonella Haemophilus* commonly characterized in both sites provide clues of the transfer of oral microbe to the intestine, even in healthy individuals [7,56]. Oral–gut translocation and ectopic intestinal colonization in a healthy human are likely to boost the establishment and maintenance of intestinal immunity. Nevertheless, under given conditions, ectopic intestinal colonization of a particular oral microbe may be involved in the pathogenesis of gastrointestinal diseases, such as IBD [7]. In turn, IBD patients also have oral manifestations that affect the composition of the oral microbiota [57].

### 3.1. Effects of IBD on Oral Microbiota

Many researchers have focused on this field to further determine whether IBD could affect the composition of the oral microbiota, as shown in the list in Table 1. Especially in recent years, the research has not only been limited to the microbial level, but has also been explored from the perspective of immunity and functional metabolism. Said et al. detected the composition of salivary microbiota of 35 patients with IBD and compared it with that of 24 healthy controls by 16S rDNA gene-based analyses; the data suggested that not species diversity, but species abundance played a more vital role in the differences in the salivary microbiota between IBD groups and healthy controls [58]. After the analysis of five dominant phyla, the results showed that compared with healthy controls, *Bacteroidetes* was notably enhanced along with a reduction in *Proteobacteria* in CD and UC groups. Further analysis of 107 genera indicated that in IBD patients the comparative increase in *Bacteroidetes* was mainly related to the rise of *Prevotella* while the decline of *Neisseria* and *Haemophilus* was largely contributed to the decrease in Proteobacteria in IBD patients. The study further showed a distinct change in salivary immunological markers of IBD patients, including elevated levels of Ig A and IL-1β with a lower lysozyme level [58,59]. However, limited by the technology of the time, Said’s analysis of biodiversity was performed considering only a limited number of diverse components. Thus, Zhang et al. analyzed the salivary microbiota of CD patients during active and remission stages of the disease and healthy controls; in addition to the same results as Said, they also indicated that *Proteobacteria*, *Pasteurellaceae*, *Alloprevotella*, *Gammaproteobacteria*, and *Desulfobulbus* were notably more abundant in healthy controls compared with CD patients during an active stage of the disease [60]. However, *Veillonellaceae*, *Negativicutes*, *Actinobacteria*, *Pedobacter*, *Salmonella*, *Prevotella*, *Bacteroidetes*, and *Bacteroidia* were enriched in the active stage [60]. They further functionally characterized the data using Phylogenetic Investigation of Communities by Reconstruction of Unobserved States (PICRUSt), which denoted in the active phase the highest enrichment among metabolic pathways such as amino sugar, nucleotide, fructose, and mannose metabolism, as well as galactose metabolism, compared with healthy controls [60]. Changes in the oral microbiota in patients with IBD are increasingly revealed by recent studies, as shown in Table 1 below [61]. In addition, a study also observed metabolic changes in IBD patients by KEGG and COG functional pathway abundance analysis, such as an increase in the metabolism of carbohydrates and energy, as well as protein processing in the ER, while the metabolism of tyrosine and genetic information processes are correspondingly reduced [61]. Moreover, to further clarify the interaction between oral microbes and inflammatory markers in IBD, this study measured the levels of white blood cell (WBC), C-reactive protein (CRP), fecal calprotectin (FC), and erythrocyte sedimentation rate (ESR) which correlate with IBD, then correlation analysis and redundancy analysis was carried out, and the results demonstrated a positive correlation between SR1 and FC and showed that TM7 positively correlated with ESR, FC, and CRP [61]. After the above research made substantive headway in the effects of IBD on the oral microbiota, and considering that oral samples are non-invasively accessible, the change in oral microbiota may potentially serve as biomarkers during the active stage of the disease.

Moreover, some IBD patients can show various oral pathological changes, with the prevalence ranging from 5% to 50% [62]. Patients with IBD, particularly CD, commonly present with oral mucosal inflammation, which includes, but is not limited to, minor aphthous lesions, mucogingivitis, and angular cheilitis; as a more detailed description, this including a coarse and messy mucosal fold, longitudinal and slit-like ulcers with a cobblestone appearance, and more serious non-caseous granuloma that protrudes from the labia and face, known as orofacial granulomatosis [63]. Although the pathogenesis of these oral manifestations is unclear, one study demonstrated that IBD-induced oral microbiota dysbiosis might also play an important role [58]. In addition, the latest prospective cohort study showed that the markers of gingivitis and periodontitis were significantly higher in patients with IBD, and perianal disease was found to be a risk factor for periodontitis by univariate analysis and logistic regression [64]. Based on the above research progress, considering that oral samples are non-invasive and accessible, oral flora, intestinal flora, and skin are more stable and change less over time, and changes in oral flora and oral mucosal lesions may have the potential to serve as biomarkers of disease activity and provide auxiliary paths for disease diagnosis [65].

### 3.2. Oral Microbiota Detected in IBD

Many previous pieces of research that have observed oral microbes in IBD patients’ guts are summarized in Table 2 and have further confirmed that certain oral microbes are strongly involved in the development of IBD.

#### 3.2.1. *Streptococcus* spp. One of the First Settlers to Reside in the Mouth and Intestines after Birth

As one of the first settlers in people’s oral cavities and intestines after birth, *Streptococcus salivarius* is assumed to play a prominent part in the establishment of immune homeostasis and the regulation of host inflammatory responses. A previous study [75] has shown that the *S. salivarius* TOVE-R strain was observed to have an inhibition effect on virulent Streptococci involved in tooth decay and pharyngitis, or pathogens involved in periodontitis. Moreover, other research [76] co-cultured human bronchial epithelial cells (16HBE14O-cells) and *Pseudomonas aeruginosa* strain PAO1, with or without *S. salivarius* K12, then monitored the responses of 16HBE14O-cells to *S. salivarius* K12 through ELISA and microarray-based analyses. This research found that *S. salivarius* K12 was able to inhibit the activation of NF-κB signaling pathways caused by *Pseudomonas aeruginosa*, suggesting its effects on the immune regulation of human epithelial cell. Given that *S. salivarius* also colonized the intestinal epithelium, this is reasonably associated with its potential to inhibit inflammatory responses and protect the intestinal epithelium, and later experiments confirmed this. Kaci et al. [77] examined NF-κB modulation stimulated by TNF-α, the pro-inflammatory cytokine in HT-29/kB-luc-E reporter cells, when four live or heat-killed strains of *S. salivarius* were present, and hypothesized that *S. salivarius* could inhibit the activation of the NF-κB pathway in human intestinal epithelial cells in vitro [77]. It is fascinating that only the live rather than heat-killed strain *S. salivarius* JIM8772 are considered downregulated. Inflammation may have a significant impact on a moderate and severe colitis mouse model, which implies the physiological and metabolic activities of bacteria are affected by the protective response while specific mechanism still needs further work to clarify [77]. Later research refined the study of the mechanism and found out that *S. salivarius* enabled the inhibition of the transcription factor PPARγ (peroxisome proliferator-activated receptor), which was associated with the degradation of the p65 subunit of the multipurpose nuclear factor NF-κB [78,79]. On the other hand, PPARγ can simultaneously reduce the successive expression of targeted metabolic genes in the intestinal epithelial cells, such as *Angptl4* (Angiopoietin like protein-4) and I-FABP whose gene products are involved in modulating lipid accumulation [78,79]. These studies more or less indicated that *S. salivarius* probably played a role in protecting intestinal health and homeostasis by impacting both host inflammatory and metabolic regulation, and displayed clinical application prospect of probiotics in future IBD treatment. Considering that the metabolic regulation has not been satisfactorily explained, this may be the direction of future research.

Regarded as “the chief criminal” of human dental caries, Streptococcus mutans have four different serotypes (c/e/f/k), among which serotype c is the most commonly seen in the oral cavity [80], while serotypes e and f have been found in endothelial cells [81]. We can isolate serotype k from the blood of infective endocarditis patients occasionally, suggesting this serotype can survive in the blood, presumably due to its lower phagocytotic capabilities and lower antigenicity [82]. Kojima et al. reported that the inspection frequency of some certain strains of *S. mutans* in patients with UC was 14.29%, which was significantly higher than that in healthy people (3.53%) [69]. Further investigating miscellaneous *S. mutans* strains in mice with dextran sodium sulfate (DSS)-induced colitis, they found that TW295, the specific strain of *S. mutans* serotype k, synthesizes collagen-binding protein (CBP) and might be considered as a latent risk in the occurrence and deterioration of UC [69]. After invading the blood from the oral cavity, TW295 escaped phagocytosis and evaded recognition by the immune cells through peculiar glucose side chains, which can survive in the bloodstream for a long time [69]. After the journey to the liver, TW295 expresses CBP to achieve adhesion and invasion of hepatocytes, which then respond to this stimulation by producing IFN-γ, subsequently leading to the release of manifold kinds of inflammatory molecule cytokines (Figure 2a,b,d). Such molecules finally reach the colon to worsen colon inflammation [69]. Interestingly, oral administration of TW295 did not aggravate colitis, which indicated that its approach is from blood circulation, rather than the digestive tract [69]. Given that only 10^4^ bacterial invasions have the ability to worsen UC, which is easily achieved by some dental procedures, including tooth brushing and extraction, we have to pay more attention [83]. Kojima et al. [84,85] further investigated the relationship between other oral streptococci and aggravation of IBD exerting a DSS-induced colitis mouse model and the results suggested that *S. sanguinis* ATCC 10,556 and TW289 led to a noticeable worsening of DSS-induced colitis induced by boosting IFN-γ secretion after intruding into the bloodstream. Unlike *S. mutans* TW295, the mechanism by which S. sanguinis strain ATCC 10,556 and TW289 possessed no CBP might function in the different pathogenic mechanism during IBD progression needs to be further explored. The findings also demonstrated that bacteria in blood probably activated the Th1 response, inducing T cells to express IFN-γ, especially CD4^+^ Th1, CD8^+^ CTL, and NK cells predominantly in the bloodstream or spleen (Figure 2c,d) [84,85]. By comparing DSS-colitis in mice with either a blockade of VEGF-A activity or the knockout of the endothelial cell-specific IFN-γ response, a recent study demonstrated that the vascular barrier could be broken down by IFN-γ by destroying VE-cadherin, an adherence junction protein, and drive DSS-induced experimental colitis [86]. In short, these studies highlighted the potential of IFN-γ antibody to inhibit the aggravation of IBD caused by various oral streptococci.

#### 3.2.2. *Fusobacterium nucleatum* as Pathogenesis of Periodontitis

As a Gram-negative, opportunistic, obligately anaerobic bacterium, *Fusobacterium nucleatum* usually colonizes the oral cavity and is always found in dental plaque, and is involved in the pathogenesis of periodontitis. As an autochthonous bacterium, although it is relatively barren to colonize the intestines of a healthy human, many researchers have suggested that *F. nucleatum* can be one of the most significant risk factors for colorectal cancer (CRC) [87]. Chen et al. examined invasive *F. nucleatum* abundance using FISH and the detection frequency of *F. nucleatum* was higher in UC tissues (51.78%) than in normal tissues (10%) [72]. To further investigate the exact mechanism of UC caused by *F. nucleatum*, through a DSS-induced colitis model that is driven by different immunological mechanisms, they found the canonical NF-κB pathway could be activated by *F. nucleatum* and pro-inflammatory cytokine expression was up-regulated, such as IL-1β, Il-6, IL-17F, and TNF-α in CRC, these findings are as similar as the results of previous researches about human gingival epithelial cells, in which they infected HGECs with *F. nucleatum*, live or heat-killed, followed by ELISA for IL-1 β, IL-6, IL-8, and IL-10, and found *F. nucleatum* induced high levels of pro-inflammatory cytokines [88,89,90]. Previous research demonstrated that bacterial peptidoglycan targeted caspase activation and recruitment domain 3 (CARD3), finally causing NF-κB activation [91]; however, here, Chen et al. demonstrated that the IL-17F/NF-κB pathway, whether in vivo or in vitro, could be activated when *F. nucleatum* targeted CARD3 through NOD2, based on the observation that *F. nucleatum* promoted the level of *NOD2* expression in NCM460 cells while *NOD2*^−/−^ inhibited the *F. nucleatum*-induced increase in CARD3 expression in the protein level [72]. In addition, Cao et al. put forward that *F. nucleatum* induced caspase activation, recruited CARD3 to activate the endoplasmic reticulum stress pathway and led to the destruction of the mucosal barrier [89]. Additional study of preventive strategies incorporating these complementary tests indicated that *F. nucleatum* damaged the integrity of intestinal epithelium (seen Box 3 for further information) and regulated the expression and distribution of the tight junction protein, zonula occludens-1, occluded to achieve increased permeability, as well as upregulated STAT3 phosphorylation, and activated the STAT3 signaling pathway both in vivo and in vitro, which can not only induce CD4^+^T cell proliferation but also led to Th1 and Th17 subset differentiation [90]. Moreover, Engevik et al. showed *F. nucleatum* could secrete nanoparticles, and outer membrane vesicles (OMVs), which are naturally secreted by Gram-negative bacteria. Applying two different media—fractionated conditioned media and *F. nucleatum* conditioned media—to incubate HT29 cells and purifying OMVs, respectively, and using Western blotting to inspect the extra downstream targets TLR4, ERK, and CREB, they discovered that OMVs contain antigenic components which are able to stimulate TLR4 to induce the downstream activation of ERK, CREB, and NF-κB, which promote proinflammation cytokine production [92]. Other studies demonstrated that it is this ability to destroy microbes that makes *F. nucleatum* more likely to exacerbate inflammation. Nonetheless, the existence of *Lachnospiraceae*, *Bacteroidetes*, and other antibiotic-depleted microorganisms can block *F. nucleatum* colonization and inflammation in turn [92,93]. In conclusion, *F. nucleatum* can damage the colorectal cells, break down the integrality of the intestinal epithelial barrier by controlling cytokine secretion (Figure 3a), and activate the inflammatory pathway and make specific T-cells proliferate and differentiate, which may exacerbate IBD and even cause CRC.

Box 3Intestinal epithelial barrier.As the protective umbrella of the gut, the intestinal epithelial barrier can resist the invasion of various intestinal flora and the infiltration of toxins. This function strongly relies on tight junctions between intestinal epithelial cells, which include, but are not limited to, tight junctions, adherent junctions, and desmosomes. The presence of these protein complexes maintains the homeostasis of the intestinal microenvironment by preventing bacteria and toxins from entering the body. There have been a lot of correlative studies on the protective mechanism behind intestinal epithelial cells and tight junctions, which will not be detailed here.

#### 3.2.3. *Porphyromonas gingivalis* as a Keystone Pathogen for Periodontal Diseases

*Porphyromonas gingivalis* is vividly described as a “keystone pathogen” for periodontal diseases; although it is not sufficient by itself to cause illness, it might contribute to the development of gingivitis or bone loss [94,95]. Considering that understanding how *P. gingivalis* modulates the immune response may promote the development of vaccines for diseases, previous studies injected *P. gingivalis* lipopolysaccharide (LPS) and ovalbumin (OVA) into the popliteal lymph node cells or splenocytes and quantification of cytokines by ELISA, and they discovered that the secretion of IL-13, IL-5, and IL-10 remarkably increased, however, IFN-γ levels were low, which implied that *P. gingivalis* LPS were capable of inducing Th-cell and T-cell to occur a semi-TH2-like response instead of TH1-type response [95,96,97]. Detecting the expression levels of cytokines, including IL-17, IL-6, TGF-β, IL-10, and transcription factors, such as RoRγt and Foxp3 by a TLR4 blocking assay, a recent study has shown that *P.*
*gingivalis* ATCC 33,277 led to the Th17/Treg ratio increasing in the colon in vivo through the TLR4-mediated signaling pathway, which aggravated DSS-induced colitis [98]. Nakajima et al. [99] compared sham-inoculated mice to C57BL/6 mice that were orally administered *P. gingivalis* (strain W83) once, and pointed out that *P. gingivalis* obviously affected the composition of gut microbiota—specifically, with an enhancement of phylum *Bacteroidetes* and a decrease in phylum Firmicutes. In addition, *P. gingivalis* decreases gut barrier function by making the tight junction protein (TJP) gene of *tjp-1* and *occluding* less expressed, and modulates the gut immune system, causing an exacerbation of gastrointestinal inflammation. Further studies found that *P. gingivalis* can secrete proteases, gingipains, which enable it to detach intestinal mucus and degrade cytosolic zonula occludens-1 (ZO-1) [100,101]. In addition, except IL-1and GM-CSF, quite a few vital pro-inflammatory mediators made by DCs and/or T cells lose their activity due to gingipains [102], which may alter the local immune microenvironment, resulting in pathogenic microorganisms subsequently colonizing or overgrowing [103] (Figure 3a). Thus, as a highly selective inhibitory peptide against gingipain, KYT-36 may have bright prospects for clinical application in treating IBD and periodontitis [104].

#### 3.2.4. *Campylobacter concisus* Co-Existing in the Oral Environment and Gut

As a Gram-negative spiral-shaped motile bacterium, *Campylobacter concisus* lives under anaerobic or microaerobic conditions and grows better in the presence of H_2_ [105,106]. Although *C. concisus* frequently settles in the oral cavity, it can also be present in the gut of some people, which has been associated with IBD [70,106]. After both cultivation and PCR detection, Kirk et al. confirmed that IBD patients had a more significant population of *C. concisus* than healthy controls [70]. Moreover, isolating *C. concisus* from the oral and fecal environments of the same patient and aligning the genomes, the results indicated few genetic differences, which suggested that intestinal isolates most likely reflect oral strain relocation [107]. Based on adherence, host cell invasion, and toxin secretion capability, together with the existence of a toxicity-related restriction-modification system, pathogenic *C. concisus* can fall into two categories: adherent and invasive *C. concisus* (AICC) and adherent and toxinogenic *C. concisus* (AToCC) [108]. After attaching to and invading host intestinal epithelial cells, the former can trigger the immune response of intestinal epithelial cells, resulting in the production and release of IL-8, IL-12, and IFN-γ. Thanks to the inhibition of the autophagy pathway, AICC earns a chance to survive within the epithelial cells [108] (Figure 3b). Unfortunately, unlike AICC, AToCC has no opportunity of staying but will be effectively cleared by autophagy after invading the epithelial cells [108]. However, Mahendran et al. found that AToCC enables the secretion of a Zot, encoded by zonula occludens toxin (*zot*) genes, and targets the tight junctions of epithelial cells, due to prolonged damage of the intestinal epithelial barrier via the Caco-2 cell model. Following this, Zot induces host cell apoptosis and the production of TNF-α and IL-8. Meanwhile, Zot has also been proven to upregulate the secretion of TNF-α in THP-1 macrophage-like cells as well as enhance the responses of THP-1 macrophage-like cells to *E. coli* K12 [109]. Furthermore, other research showed that *C. concisus* ATCC 33,237 had the capability to upregulate CD11b, which is the neutrophil adherence molecule, and enhance the oxidative burst response, which led to activating the innate immune system [110] (Figure 3b). Interestingly, new research suggested that of *C. concisus* possessed better motility in microaerophilic conditions than in an anaerobic environment. By inflammatory processes such as neutrophil oxidative burst, the oxygen levels in the inflamed gut potentially increased, activating the ability of *C. concisus* to cause epithelial damage [111]. This ‘vicious cycle’ broadens the way we look for the pathogenesis and treatment of IBD.

#### 3.2.5. *Klebsiella pneumoniae* Located in the Mucous Layer

*Klebsiella pneumoniae* is a Gram-negative pathogen, settling in the mucosal layer of mammals, including the oral cavity, upper respiratory tract, and intestinal tract [112]. As a common explanation for why it is easy for hospitalized patients to get antimicrobial-resistant opportunistic infections, *K. pneumoniae* has won wide attention [113]. In addition to causing pneumonia [112], further research has linked it to IBD. Lee et al. explored the roles of *K. pneumoniae* in TNBS-induced colitis in mice, and finally found that *K. pneumoniae* produced β-glucuronidase and LPS to induce murine peritoneal macrophages to produce NO and COX-2. LPS isolated from *K. pneumoniae* activated NF-κB, which is the representative transcription factor in IBD, enhancing potent pro-inflammatory stimulators, which contained IL-1, IL-6, and TNF-α production and secretion levels in mucosal macrophages. In addition, the lipid peroxide that participates in IBD pathogenesis is increased due to *K. pneumoniae*. However, tight junction associated proteins, claudin-1, ZO-1, and occluding expressions were at a low level and this made bacterial invasion and penetration into the colonic mucosa easier [114]. Recently, a study made by Atarashi et al. also showed that *K. pneumoniae* that stem from the oral microbiota were able to colonize in the gut, potently inducing chronic intestinal inflammation [115]. By transplanting saliva samples from two CD patients into a germ-free mouse, they found that T helper 1 (TH1) cell levels in epithelial lamina were significantly increased in one of the mice, but only a small portion of the microbial species detected in the fecal microbes of the mice are homologous to salivary, suggesting that a small part of oral microbes could have the chance of immigrating and colonizing intestine to induce Th1 cell-mediated inflammation [115]. In order to further determine which Th1 cell-inducing bacteria it was, they separately cultured mice with eight major bacteria isolated from the feces of mice and finally found that a strain of *K. pneumoniae* (Kp-2H7) could induce the expression of Th1 cells. Further study indicated that only under certain circumstances, for instance, under antibiotic-induced microbiota perturbation or in IL-10-deficient mice, oral Kp-2H7 induce intestinal colonization and pathogenic inflammation [115]. Moreover, they showed that *Klebsiella* antigen-specific TH1 responses, which were maintained in the way of an IFN-γ-mediated feedforward loop, occurred via stimulation of the innate immune receptor TLR4 [115]. In addition, the presentation of bacterial surface antigens required, for example, OmpX, which is presented by one of the primary cells for antigen presentation in the intestine—Batf3-dependent CD11b^−^CD103^+^ DCs. In addition, after the activation of TLR4 signaling epithelial cells can release IL-18 to amplify the TH1 response in turn [115,116,117]. In addition to the fact that *K. pneumoniae* itself can directly translocate to the gut to trigger inflammation, Kitamoto et al. [118] found that periodontitis caused by *K. pneumoniae* resulted in oral pathobiont-reactive Th17 cells being generated in the mouth, which can be transferred to the inflamed gut along the lymph node to induce colitis (Figure 3c).

#### 3.2.6. TM7 Associated with IBD but Not to Be Ignored

As a very small component of the oral microbiota, TM7 has been found in human subgingival plaque. Previous research have noted the significant role which TM7 division plays in the early stages of the inflammatory mucosal disease [23,24]. Comparing the mucosal microbes of inflammatory sites in patients who are in active CD and UC with non-IBD controls, researchers detected CD patients had a higher diversity of TM7 phylotypes than both UC patients and non-IBD controls, and the diversity of TM7 in UC was approximately similar to that of controls. Due to phylogenetic analysis, the data hinted that the majority of TM7 clone sequences in the samples were close relatives of oral clones, and the clone detected in UC patients were 99% similar to oral clone I025, which was considered as a presumed certain pathogen for oral disease [24,119]. It is worth noting that TM7 does not directly cause inflammation. Nonetheless, it may modulate the community structure and function of the oral microorganism by affecting the physiology of the bacterial host, restraining the host’s growth, or directly killing it to affect its relative abundance, thus causing some oral diseases [28,119]. Overall, the evidence, depending on the research, suggests that TM7 is highly likely to be related to IBD, although the pathophysiology of bacterial triggers may differ in UC and CD. However, the underlying mechanisms of IBD caused by TM7 require further investigation.

#### 3.2.7. *Candida albicans* as an Important Pathogenic Fungus

In addition to the bacteria mentioned above, with the widespread use of antibiotics and immunosuppressive drugs and the increasing number of patients with acquired immune deficiency syndrome, fungus-induced diseases are attracting increasing attention. As an important pathogenic fungus in the human body, *Candida albicans* usually reside in the oral cavity, upper respiratory tract, intestinal tract, and vagina of normal people. Under normal circumstances, a small number of *C. albicans* will not cause disease, but when the body’s immune system is compromised or dysbiosis occurs, it will multiply in large numbers and change from the yeast phase to mycelium phase, causing disease.

A study collected mouth swabs and fecal specimens from 41 families of CD patients and 14 control families for quantification of *C. albicans* as well as serum samples for the test of anti-Saccharomyces cerevisiae antibodies (ASCA), which is an important marker of CD, and the results showed that *C. albicans* was colonized more frequently and more severely in the intestinal tract of patients with CD and their first-degree healthy relatives (HRs) than in the control group. In HRs, ASCA could derive from a changed immune response to *C. albicans* [74]. Other studies have shown that the presence of ASCA may be found in approximately 60% of CD patients. Several studies have noted ASCA expression to be almost 95% specific for CD despite modest sensitivity, which suggests its exciting future as a diagnostic marker for CD. In addition, ASCA serology may also be relevant to disease behavior, location, and increased risk for early surgery [120]. For the relationship between *C. albicans* and IBD, some studies mentioned that exogenous *C. albicans* promotes the growth of *E. coli*, which is thought to be a significant reason for the deterioration of *C. albicans*-associated colitis because correlations between *E. coli* and relative fungal abundance were clearly decreased in colistin-treated mice compared to vancomycin-treated and control mice [121,122]. Besides, Choteau et al. explored how toll-like receptor (TLR) deficiency affected changes in inflammatory parameters related to *C. albicans* colonization and DSS-induced acute colitis by comparing wild-type, *TLR1*^−/−^, *TLR2*^−/−^, and *TLR6*^−/−^ mice. The results indicated that the overgrowth of *E. coli* and *C. albicans* was promoted by DSS-induced colitis in the gut. Furthermore, the deletion of TLR1 and TLR2 accelerated intestinal inflammation which was induced by *C. albicans* colonization by way of the strong upregulation of TNF, IL-1β, and IL17A, which brought about the injury of the colon and the death of the mouse. On the contrary, decreasing the production of IL10 and promoting *C. albicans* elimination is the specific mechanism of intestinal inflammation concerned with the deletion of TLR6 [123]. Based on the fact that paeonol (PAE) could effectively alleviate systemic inflammation in the DSS-induced UC model with *C. albicans* via downregulating serum β-glucan, ASCA, pro-inflammatory cytokines and increasing the expressions of Dectin-1, TLR2, and TLR4, Ge et al. suggested PAE may be regarded as a candidate for the treatment of UC patients whose pathogenesis is related to fungal dysbiosis, and has an outstanding possibility to activate the Dectin-1/NF-κB pathway cooperating with TLR2 and TLR4. Besides the inhibition or elimination of *E. coli*, PAE is also involved in suppressing the production of pro-inflammatory cytokines induced by LPS, such as TNF-α, IL-1β, IL-6, and upregulating the level of IL-10, which is the anti-inflammatory cytokine in macrophages [124].

The treatment of immune-related diseases caused by microorganisms has come into full swing, and exploring the relationship between oral microbes and IBD may provide a novel strategy for the anti-inflammatory treatment of IBD. The use of probiotics or antibiotics to inhibit specific oral microorganisms is expected to give a great impetus to treating IBD in the future. In pursuit of this, further research on the underlying pathogenesis is needed.

## 4. Possible Pathways of Ectopic Gut Colonization by Oral Microbiota

Emerging evidence suggests oral bacteria are likely to have a direct immunomodulatory bearing on the intestinal mucosa, but how migrating to the gut mucosa is mediated remains undiscovered. An intermittent yet persistent migration of oral bacteria by hematogenous or via the enteral route (Figure 1 and Figure 2a) after mastication and personal oral hygiene may reach the liver and other sites, as we previously reviewed. As suggested by previous findings, except in pathological conditions, such as periodontitis [125], routine daily dental activity, including invasive dental treatments with tooth extraction and scaling, hard mastication [85] and bleeding during brushing [126], flossing and interdental brushing [127], and even just routine chewing [126] in daily life, could cause oral mechanical injuries [126,127,128], which give oral microbes, such as *Porphyromonas gingivalis*, *Streptococcus salivarius* and *Streptococcus sanguinis*, a chance to spread into the bloodstream. Additionally, metabolic products or toxins of oral microbes can diffuse into the blood, which in turn causes chronic systemic inflammation and probably contributes to the occurrence and progression of IBD [128]. Furthermore, the mode of hematogenous migration can also be indirectly achieved by invasion and survival inside immune cells, like DCs and macrophages in the blood [129]. Another mode is by enteral spreading. Although around 1.5 L of saliva, which can deliver enzymes, effector cytokines, countless oral microbes, as well as various inflammatory cells, to the gut, is swallowed every day [130], few of them reach and settle in a healthy intestine due to the presence of gastric acidity and intestinal mucosal barrier [7,131]. Gastric acid is one of the largest stumbling blocks for oral microbes to move from the oral cavity to the gut, because several oral bacteria tend to be sensitive to gastric acid. In this way, the enteral route maybe not very efficient for normal people. However, in patients with a long drug history of proton pump inhibitors, subtotal gastrectomy, or *Helicobacter pylori* infection, the enteral route is noteworthy due to their reduced gastric acid. In addition to these two main routes, a recent study compared the development of three *C. concisus* strains in BHI agar motility plate under anaerobic and microaerobic environments and put forward the idea that oxygen may alter the physiological role of *C. concisus*, promoting the dissemination and increasing mucosal adherence to facilitate colonization [111].

It is worth noting that in immune-compromised individuals, due to immunomodulating drugs targeting these immune-suppressive mechanisms and systemic antibiotic administration, in contrast to gut communities, the migration of oral microbes into the gut is more likely to cause IBD [7,117].

## 5. Conclusions and Future Perspective

At present, most of the microbial studies on IBD are based on the intestinal microbiota, but many recent studies have pointed out that the influence of the oral microbiota on the occurrence and exacerbation of IBD may have been underestimated. Considering that IBD can induce some changes in the oral microbiota composition, as well as the fact that saliva samples have the advantages of being easy to obtain and non-invasive, we are hopeful that the detection of the oral microbiota may have potential in the diagnosis of IBD. As microbe-based therapies are becoming more various and promising, summarizing the mechanisms of oral microbiota leading to or exacerbating IBD will probably inspire new ideas for the anti-inflammatory therapy of IBD. Although we have summarized the relationship between oral microbiota and IBD in detail, we cannot describe them one by one because of their large number; the existing data hint that there is a close relationship between oral microbiota and IBD. In the future, the use of probiotics or antibiotics to inhibit specific oral microbes is expected to inject new impetus into the treatment of IBD. To achieve it, further research on its pathogenesis and its exact mechanism needs to be undertaken. At the same time, in the daily care of IBD patients, the maintenance of the oral health environment should be improved to prevent the aggravation of the disease.

## Figures and Tables

**Figure 1 microorganisms-10-01868-f001:**
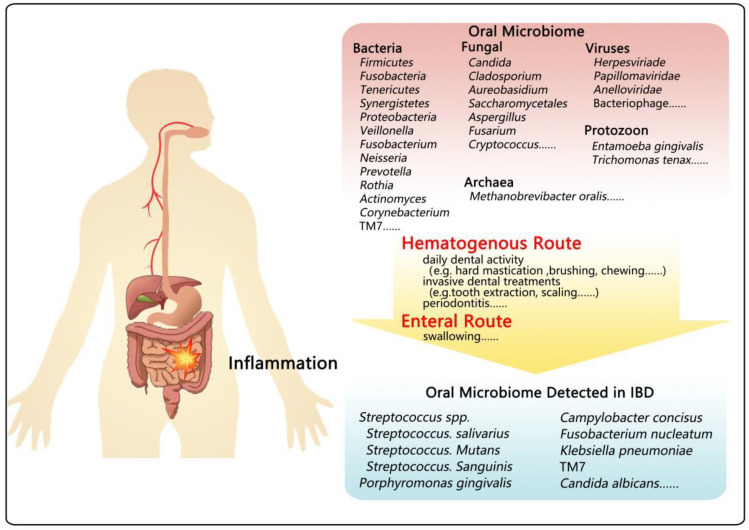
The composition of oral microbiota detected in IBD subjects. Oral microbiota is composed of multiple members, including bacteria, fungi, viruses, protozoa, and archaea, some of which can translocate to the gut through the bloodstream or enteral spreading, resulting in or aggravating IBD.

**Figure 2 microorganisms-10-01868-f002:**
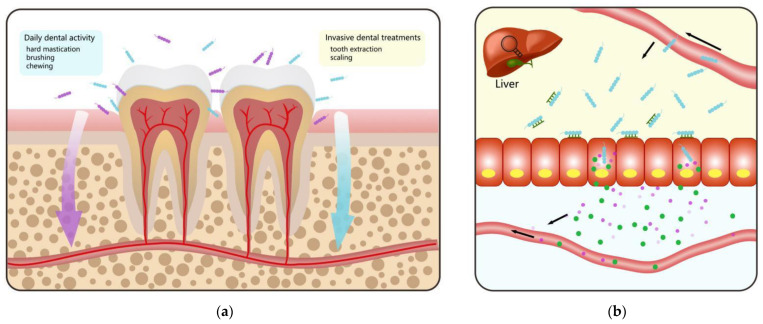
Pathogenesis of *Streptococcus* spp. in IBD. (**a**) *S. mutans* and *S. sanguinis* can enter the bloodstream from the oral cavity through daily dental activities or invasive dental treatments. (**b**) After circulating to the liver, *S. mutans* TW295 can express CBP to achieve adhesion and invasion of hepatocytes, which then respond to this stimulation by producing IFN-γ and subsequently leading to the release of all sorts of inflammatory molecules cytokines. (**c**) *S. sanguinis* ATCC 10,556 and TW289 may activate the Th1 response, inducing T cells to express IFN-γ, especially CD4^+^ Th1, CD8^+^ CTL, and NK cells predominantly in the bloodstream or spleen. (**d**) Inflammatory molecules and IFN-γ produced by both bacteria travel through the bloodstream to the colon, aggravating the inflammation of colitis.

**Figure 3 microorganisms-10-01868-f003:**
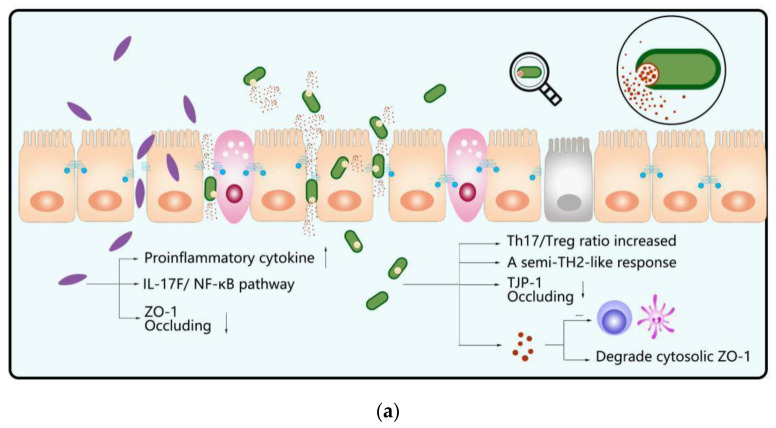
Specific oral pathogenic microorganisms and underlying mechanism patterns in IBD. (**a**) In addition to activating the IL-17F/NF-κB pathway, *F. nucleatum* can also damage the integrity of intestinal epithelium by regulating the expression and distribution of tight junction proteins zonula occludens-1 and occluding, and promoting proinflammatory cytokine production to exacerbate IBD. *P. gingivalis* can secrete proteases, gingipains, detaching intestinal mucus by degrading ZO-1, inactivating quite a few vital pro-inflammatory mediators made by DCs and/or T cells, which may alter the local immune microenvironment. In addition, *P. gingivalis* itself can induce a semi-TH2-like response, increasing the Th17/Treg ratio in the colon as well as impacting the composition of gut microbiota, and decrease gut-barrier function through reducing the expression of TJP-1 and occluding. (**b**) AICC attaches to and invades host intestinal epithelial cells, triggering the host cell to produce IL-8, IL-12, and IFN-γto maintain its own survival by inhibiting the autophagy pathway. AToCC enables the secretion of Zot, which targets the tight junctions of epithelial cells to damage the intestinal epithelial barrier, induces host cell apoptosis and the epithelial production of TNF-α and IL-8, and upregulates the production of TNF-α in THP-1 macrophage-like cells. In addition, a specific strain of *C. concisus* enables the upregulation of the neutrophil-adherence molecule CD11b. (**c**) *K. pneumoniae*, a Gram-negative bacterium, can reside in the gut, downregulate the expression of TJP, claudin-1, and ZO-1, and increase the activity of mucosal macrophages to secrete proinflammatory cytokines, like IL-1, IL-6, and TNF-α, to trigger gut inflammation. *K. pneumoniae* activates Th1 response via stimulation of TLR-4 and presentation of OmpX by DC, and further amplifies Th1 response by activating epithelial cells to produce IL-18 through TLR-4. Moreover, periodontitis caused by *K. pneumoniae* results in oral pathobiont-reactive Th17 cells being generated in the oral cavity, which can be transferred to the inflamed gut along the lymph node to induce colitis.

**Table 1 microorganisms-10-01868-t001:** Oral microbiota changes in inflammatory bowel disease.

Disease	Subjects	Specimen	Oral Microbiota with Increased Abundance	Oral Microbiome with Decreased Abundance	Detection Method	Refs
IBD	HCs (*n* = 43) CD (*n* = 40) UC (*n* = 31)	Tongue and buccal mucosal brushings	*Bacteroidetes* *Spirochaetes* *Synergistetes*	*Fusobacteria* *Firmicutes*	HOMIM	[66]
Adult IBD with untreated chronic periodontitis	HCs (*n* = 15) CD (*n* = 15) UC (*n* = 15)	Gingivitis and periodontitis sites samples	*Prevotella Melaninogenica* *Staphylococcus aureus* *Streptococcus anginosus* and *Streptococcus mutans* in CD *Staphylococcus aureus* *Peptostreptococcus anaerobius* in UC		checkerboard DNA-DNA hybridization technique	[67]
Adult IBD	HC (*n* = 24) CD (*n* = 21) UC (*n* = 14)	Saliva	*Prevotella* *Veillonella* Bacteroidetes	*Streptococcus* *Haemophilus* Proteobacteria *Neisseria* in CD *Gemella* in CD	PCR	[58]
Pediatric CD	discovery cohort HC (*n* = 46) CD (*n* = 35) validation cohort HC (*n* = 31) CD (*n* = 44)	Subgingival plaque samples	*Capnocytophaga* *Rothia* TM7		PCR	[68]
Adult IBD	HC (*n* = 25) CD (*n* = 13) UC (*n* = 54)	Saliva	*Veillonellaceae* in CD Streptococcaceae and Enterobacteriaceae in UC	Neisseriaceae and Haemophilus in CD Lachnospiraceae and [Prevotella] in UC	PCR	[57]
Adult IBD	HC (*n* = 8) CD (*n* = 12) UC (*n* = 10)	Saliva	*Saccharibacteria* (TM7) *Absconditabacteria* (SR1) *Leptotrichia* *Prevotella* *Bulleidia* *Atopobium*	*Streptococcus* *Rothia*	PCR	[61]

IBD inflammatory bowel diseases, CD: Crohn’s disease, UC: ulcerative colitis, HCs: healthy controls, PCR: polymerase chain reaction, HOMIM: a custom-designed, 16S rRNA-based oligonucleotide reverse capture microarray.

**Table 2 microorganisms-10-01868-t002:** Detection rate of oral microbes in inflammatory bowel disease.

Oral Microbes	Disease	Specimen	Detection Rate		*p*	Detection Method	Refs
			IBD	HCs			
Streptococcus. mutans	UC	Oral sample	14.29%	3.53%	0.0012	-	[69]
Campylobacter concisus	IBD	gut mucosal biopsies from adult	49% (121/245)	36% (6/182)	0.008	PCR	[70]
Campylobacter concisus	CD	Intestinal biopsy specimens	51% (17/33)	2% (1/52)	<0.001	PCR	[71]
Fusobacterium nucleatum	UC	Biopsies samples	51.78%	10%	0.017	FISH analysis	[72]
Klebsiella	UC	Biopsies samples	31.0% (9/29)	-	-	PCR	[73]
	CD		21.4(3/14)	-	-		
TM7	CD	Biopsies samples	51.5%	51.6%	<0.01	PCR	[24]
	UC		52.4%	-			
Candida albicans	CD	Stool samples	43.9% (47/129)	22% (13/76)	<0.05	Chromogenic medium	[74]

IBD: inflammatory bowel diseases, CD: Crohn’s disease, UC: ulcerative colitis, HCs: healthy controls, PCR: polymerase chain reaction, FISH: fluorescence in situ hybridization.

## Data Availability

Not applicable.

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
