# Peer review of "Insight into the Relationship between Oral Microbiota and the Inflammatory Bowel Disease"

_microorganisms, 2022, doi:10.3390/microorganisms10091868_

Round 1

Reviewer 1 Report (Previous Reviewer 2)

no more problem except to check the plagiarism score. Long sentences with only 1 reference

Author Response

Point 1: no more problem except to check the plagiarism score. Long sentences with only 1 reference.

Response 1: Thank you very much for your positive comments on this review. Before submitting the manuscript, we have carried out the duplicate checking of papers, and after submitting the manuscript, the editorial office has also carried out the corresponding repetition rate query. In addition, according to your great suggestion, we have also modified the long sentence with only one reference, please see page 3 line 93-103, page 7 line 278-299, page 9 line 345-353, page 10 line 354-359, page 10 line 372-384, page 10 line 395, page 15 line 570-585. Thank you again for your important tips.

Reviewer 2 Report (Previous Reviewer 1)

I suggested that the authors provide a stronger emphasis to “the linkage of changes in the paediatric microbiome to the occurrence of adult IBD.” The short introduction in lines 124-126 is inadequate to this important concept.  

The standard of English has improved but assistance from a native English-speaking scientist or clinician is still necessary.

References 61-86, 94-98 and 126-130 contain no citations more recent than 2015; most of these references must be replaced by current research.

Of the recent references suggested by this reviewer, only one seems to have been included. Why?

Author Response

Point 1: I suggested that the authors provide a stronger emphasis to “the linkage of changes in the paediatric microbiome to the occurrence of adult IBD.” The short introduction in lines 124-126 is inadequate to this important concept.

Response 1: Thank you very much for your constructive and important comments. According to your suggestion, we have supplemented the relevant content, please see page 3 line 111-122. Thanks again.

Point 2: The standard of English has improved but assistance from a native English-speaking scientist or clinician is still necessary.

Response 2: Thank you very much for your generous comments. According to your advice, we have made comprehensive and accurate modifications and improvements to the language expressions, grammatical structures and other problems of language in the manuscript again. We believe that the modifications will greatly ameliorate the accuracy of language expressions so as to improve the readability of our manuscript. All modified areas are highlighted in bright blue.

Point 3: References 61-86, 94-98 and 126-130 contain no citations more recent than 2015; most of these references must be replaced by current research.

Response 3: Thank you very much for your valuable advice. According to your suggestion, we have reviewed a lot of relevant literature, and then updated the literature you mentioned above. Please see the literature[19],[20],[63],[65],[66],[80],[82],[85],[95],[97], and [125-127]. Thank you again for your important tips.

Point 4: Of the recent references suggested by this reviewer, only one seems to have been included. Why?

Response 4: Thank you for your objective and valuable comments. According to the comments of the reviewer last time, we have carefully read the references he put forward. However, since most of the articles provided by reviewers are review types, we have revised the articles and cited the original literature after careful reading. Thank you again for your important tips.

Reviewer 3 Report (New Reviewer)

Comments to the Authors of manuscript number: microorganism-1903072 entitled “Insight into the relationship between oral microbiota and the Inflammatory Bowel Disease”.

The review seems to be very interesting, however there is a very big problem. Although I am not Anglophone, the text is unclear due to the language. It should be corrected substantially. After the needed correction is can be published.

1. Abstract: the abbreviation should be provided in the first sentence.

2. L 27 – general

3. L 29- showed

4. L 54- the current knowledge

5. L 56-57 and treatment of IBD

6. L 60 – how it can summarize the composition? Is should be rephrased

7. L 69-71- the phrase is hard to understand e.g. “microenvironment is perturbation’?

8. L 70 – result in

9. L 70-71- “result in oral ailments”???

10. I cannot read further because each earlier corrected sentence is wrong.

Author Response

Point 1: Comments to the Authors of manuscript number: microorganism-1903072 entitled “Insight into the relationship between oral microbiota and the Inflammatory Bowel Disease”.

The review seems to be very interesting, however there is a very big problem. Although I am not Anglophone, the text is unclear due to the language. It should be corrected substantially. After the needed correction is can be published.

Response 1: Thank you very much for your comments on this review. According to your suggestion, we sincerely invited Dr. Xin Tao from the Department of Applied Linguistics, Capital Medical University, to make comprehensive and accurate modifications and improvements to the language expressions, grammatical structures and other problems of language in the manuscript. We believe that the modifications will greatly ameliorate the accuracy of language expressions so as to improve the readability of our manuscript. The revised part has been marked up using the bright blue so that the modified portion can be seen more clearly.

Point 2: Abstract: the abbreviation should be provided in the first sentence.

Response 2: Thank you very much for your advice. According to your suggestion, we have made corresponding modifications. Please see page 1 line 22, page 1 line 28, page 1 line 29, page 1 line 31, page 1 line 32-33, and page 1 line 35. Thanks again.

Point 3: L 27 – general

Response 3: Thank you very much for your kind advice on the improper expression. According to your suggestion, we have made corresponding modifications. Please see page 1 line 27. Thanks very much.

Point 4: L 29- showed

Response 4: Thank you very much for valuable advice. According to your suggestion, we have made corresponding modifications. Please see page 1 line 28. Thanks again.

Point 5: L 54- the current knowledge

Response 5: Thank you very much for your kind detailed comment. According to your suggestion, we have made corresponding modifications. Please see page 2 line 51. Thanks.

Point 6: L 56-57 and treatment of IBD

Response 6: Thank you for your kind detailed comment. According to your suggestion, we have made corresponding modifications. Please see page 2 line 53. Thanks again.

Point 7: L 60 – how it can summarize the composition? Is should be rephrased

Response 7: Thank you very much for your detailed and valuable comment. According to your suggestion, we have made corresponding modifications. Please see page 2 line 55. Thanks again.

Point 8: L 69-71- the phrase is hard to understand e.g. “microenvironment is perturbation’?

Response 8: Thank you very much for your valuable comment. According to your suggestion, we have made corresponding modifications. Please see page 2 line 65. Thanks again.

Point 9: L 70 – result in

Response 9: Thank you very much for your detailed advice. According to your suggestion, we have made corresponding modifications. Please see page 2 line 65. Thanks again.

Point 10: L 70-71- “result in oral ailments”???

Response 10: Thank you very much for your important advice. According to your suggestion, we have made corresponding modifications. Please see page 2 line 65. Thanks.

Point 11: I cannot read further because each earlier corrected sentence is wrong.

Response 11: Thank you very much for your generous suggestions. According to your suggestions, we sincerely invited Dr. Tao Xun from the Department of Applied Linguistics of Capital Medical University to polish the grammar and expression of this article thoroughly, so that this article can be more readable. The modified part has been marked in bright blue so as to make them highlighted. Thank you again for your important tips.

Round 2

Reviewer 2 Report (Previous Reviewer 1)

The authors have made appropriate changes to the manuscript. 

Reviewer 3 Report (New Reviewer)

I still have some issues with the English language and typos in this mauscript. Just a one characteristic example - figure 3A. ZO-1 and TJP-1 are aliases of the same protein - Zonula occludens-1. Further, "occliuding" is another typo, a correct name for the protein is "occlodin". Please revise the manuscript once again and correct all mistakes. Correct language and the absence of errors make up a significant part of the success of any paper, especially review work.

This manuscript is a resubmission of an earlier submission. The following is a list of the peer review reports and author responses from that submission.

Round 1

Reviewer 1 Report

This review covers an interesting topic which is well worth a detailed review. 

The standard of English of overall satisfactory but assistance from a native English speaker with extensive scientific or clinical experience would help the readers.  

Lines 301-312: Delete as this statement is correct but not relevant to the topic of the review.

One topic that could have a stronger emphasis is the linkage of changes in the paediatric microbiome to the occurrence of adult IBD. This may need a short introduction of the fetal origin of adult disease, especially cardiovascular disease (PMID: 32115330), metabolic diseases (PMID: 33270534) and neuropsychiatric diseases (DOI: 10.1111/apa.15766).

The text emphasises the presence of different bacteria but isn’t the density more important in initiation of disease? 

Section 4 appears to emphasise periodontitis rather than the connection to IBD.

Line 816: Is polishing the images enough to provide intellectual input into the topic of a review to justify authorship?

For a review on such a quickly evolving topic, the proportion of references that are pre-2015 is surprising. The authors should strongly consider updating these references or deleting them. 

Further, PubMed gives the following recent references which do not seem to have been cited:

DOI: 10.3389/fped.2020.620254; DOI: 10.3389/fimmu.2021.620124; DOI: 10.3389/fmicb.2018.01136 (the key points of this review do not seem to be in the text); DOI: 10.3389/fmed.2021.723719; DOI: 10.1172/jci.insight.148543. 

Reviewer 2 Report

Comments:

2. Oral microbiota: an overview : If you are talking about an overview, you should address the methodology used

2.1. Characteristics of the composition of the oral microbiota. This chapter is far too generalized. You don't talk about dysbiosis, symbiosis, etc... 

Figure. 1 The options for choosing the bacteria of the oral microbiome are questionable. As is the list proposed in the IBDs

Bloodstream. It is not clear. How for example the "daily dental activity" impacts this route of contamination. You minimize everything that is gingival bleeding, periodontal diseases.

Similarly, you do not address the problem of the interdental space, where recent work underlines the dysbiosis of the microbiota, even in young adults.

2.1.1. Characteristics of the TM7x.  What does TM7x have to do with this point in your article ?

2.2.1. Influence of metabolites-derived oral microbes in cardiovascular health . What does this have to do with your topic?

The oral microbial composition in saliva of 21 caries-free children and 20 caries-infected children were characterized and compared by the single-molecule real-time DNA 180 sequencing system, and the result showed that Prevotella spp., Lactobacillus spp., Dialister 181 spp. as well as Filifactor spp. were probably associated with the pathogenesis and progres- 182 sion of dental caries[40]. What does this have to do with your topic?

Recent studies investigated the bronchoalveolar lavage fluid of patients with COVID-19 via metagenomic next-generation sequencing and found that aside from SARS-CoVs, some oral microbiota could also be characterized, such as Capno- 200 cytophaga gingivalis, Veillonella parvula and Prevotella melaninogenica[44, 45], indicating that 201 coinfection of the oral microbiota with SARS-COV-2 can occur in the lungs of patients 202 with COVID-19[46] What does this have to do with your topic?

2.3 Factors affecting oral microbiota. You are prioritizing some factors, the list is not exhaustive I don't see exactly what the point of this chapter is with respect to your initial objective. 

3. IBD: an overview . Is it really useful to develop a whole chapter on this subject which is not related to your objective and which can be found in a more precise and developed way in any quality reference article?

Your title is: Insight into the relationship between oral microbiota and the Inflammatory Bowel Disease. However, you start with 4.1. Effects of IBD on oral microbiota . This is not the subject. Or change your title

Table 1. Oral microbiota changes in inflammatory bowel disease. What methodologies did you use to identify these references? Is the list exhaustive?

4.2 Oral microbiota detected in IBD: Yes, this is the topic. What methodologies did you use to identify these references 

4.2.1. Streptococcus spp. Please can you explain your choice to develop certain bacteria 4.2.2. Fusobacterium nucleatum ,Pg, Campylobacter concisus , 4.2.5. Klebsiella pneumoniae , 4.2.6. TM7 and 4.2.7. Candida albicans 

4.2.2.1. Intestinal epithelial barrier : this section is only related to Fusobacterium nucleatum. Normal ?

Possible pathways of ectopic gut colonization by oral microbiota. Yes 

An intermittent yet persistent migration of oral bacteria - by hematogenous 726 (tooth brushing and flossing) . No. According to you, it is because we brush our teeth or use flossing that there is bleeding. Please review. You do not insist enough on the red and orange Socransky complexes as well as the specificity of the interdental space and beyond the use of interdental brushes

6. Current and novel treatments for IBD: Off topic. The question is: How can we contribute to maintain a symbiosis of the different microbiota of the oral cavity.

However, considering the critical role of  oral microbiota in the pathogenesis of IBD, it suggests that the use of probiotics or antibiotics to regulate the composition of oral microbiota and eliminate specific pathogens, with or without other treatments, is likely to improve the symptoms and disease progression. What does this do at the end of the chapter, without reference. This is probably a point that should be addressed, but not in this way

L 805 : more and more recent studies have pointed out that the oral microbiota may have an underestimated impact on the occurrence and exacerbation of IBD. No, your article, which is not a meta-analysis or an overview, does not allow you to write that

Conclusion:

This submission, which at first reading appears professional, is poorly structured, with a confusion of objectives, chapters that do not provide any added value and present gaps, particularly on the problem of periodontal conditions. Methodological problems are sullied: We are not in the framework of a meta-analysis or even an overview of the literature, which makes the authors take options or decisions that may prove to be questionable and not based on evidence.

My advice is to refocus, to synthesize this article with respect to the proposed objective by taking into account the observations made.